# Dielectric Properties and Spectral Characteristics of Photocatalytic Constant of TiO_2_ Nanoparticles Doped with Cobalt

**DOI:** 10.3390/nano11102519

**Published:** 2021-09-27

**Authors:** Valentin G. Bessergenev, José F. Mariano, Maria Conceição Mateus, João P. Lourenço, Adwaa Ahmed, Martin Hantusch, Eberhard Burkel, Ana Maria Botelho do Rego

**Affiliations:** 1Centre of Marine Sciences (CCMAR), Campus de Gambelas, Universidade do Algarve, FCT, 8005-139 Faro, Portugal; 2CeFEMA, Campus de Gambelas, Universidade do Algarve, FCT, 8005-139 Faro, Portugal; jmariano@ualg.pt; 3Centro de Investigação em Química do Algarve (CIQA), Campus de Gambelas, Universidade do Algarve, FCT, 8005-139 Faro, Portugal; mcdamateus@gmail.com (M.C.M.); jlouren@ualg.pt (J.P.L.); 4Centro de Química Estrutural, Instituto Superior Técnico, Universidade de Lisboa, Av. Rovisco Pais, 1096-001 Lisbon, Portugal; 5Institute of Physics, University of Rostock, 18055 Rostock, Germany; adwaa.ahmed@uni-rostock.de (A.A.); eberhard.burkel@uni-rostock.de (E.B.); 6IFW Dresden, Helmholtzstraße 20, 01069 Dresden, Germany; m.hantusch@ifw-dresden.de; 7CQFM, Instituto Superior Técnico, Universidade de Lisboa, 1049-001 Lisbon, Portugal; amrego@tecnico.ulisboa.pt

**Keywords:** photocatalysis, nanomaterials, dielectric properties, titanium dioxide

## Abstract

Dielectric properties and spectral dependence of the photocatalytic constant of Co doped P25 Degussa powder were studied. Doping of TiO_2_ matrix with cobalt was achieved by precipitation method using of Tris(diethylditiocarbamate)Co(III) precursor (CoDtc–Co[(C_2_H_5_)_2_NCS_2_]_3_). Five different Co contents with nominal Co/Ti atomic ratios of 0.005, 0.01, 0.02, 0.05 and 0.10 were chosen. Along with TiO_2_:Co samples, a few samples of nanopowders prepared by Sol-Gel method were also studied. As it follows from XPS and NMR studies, there is a concentration limit (TiO_2_:0.1Co) where cobalt atoms can be uniformly distributed across the TiO_2_ matrix before metallic clusters start to form. It was also shown that CoTiO_3_ phases are formed during annealing at high temperatures. From the temperature dependence of the dielectric constant it can be concluded that the relaxation processes still take place even at temperatures below 400 °C and that oxygen defect Ti–O octahedron reorientation take place at higher temperatures. The spectral dependency of the photocatalytic constant reveals the presence of some electronic states inside the energy gap of TiO_2_ for all nanopowdered samples.

## 1. Introduction

As it was shown [1,2,3,4], additional features of titanium dioxide, that is widely used as photocatalyst and also has potential applications in photovoltaic and spintronic devices, can be achieved by its doping with 3d-transition metals. Due to high dielectric constant titanium dioxide can be used as an isolator component in electronic devices [5], or can be combined with organic–inorganic composites [6,7]. One of the points of interest is the modification of the electronic states by creating new mid-gap states in TiO_2_. In that case, additional energy states of 3d transition metal are formed inside the original TiO_2_ band gap, thus effectively reducing band gap [8]. Doping TiO_2_ with 3d metals occurs normally by heterovalent substitution. Several possible oxidation states e.g., Co^2+^, Co^3+^ and Co^4+^ and several types of coordination, including octahedral can be realized when TiO_2_ doped with cobalt. The complexity in the spin state is due to crystal field splitting of the 3d energy level of the cobalt ion in oxygen neighborhood [9]. As it was recently shown [10], the magnetic moment of the Co atom in TiO_2_ matrix depends on the cobalt content. Low Co concentration, namely, 0,5%, 1% and 2%, shows magnetization of approximately 2.5 μB/Co while for higher concentration of Co, 5% or 10%, the magnetic moment is approximately 1.3 μB/Co. This means that both oxidation states of cobalt Co^3+^ (S = 2) and Co^2+^ (S = 3/2) can be realized [10]. It is supposed that in TiO_2_:Co, cobalt occupied the Ti position inside the octahedral, where titanium is usually in Ti^4+^ or Ti^3+^ state. As it was shown [10] the different spin states can be easily realized by varying the temperature or the pressure or by applying an external magnetic field. It is due to the fact that the crystal field changes depend on the Co–O bond length and the Co–O–Co bond angle [10]. So, the magnetic properties as well as for photocatalytic activity in oxygen reduced oxides doped by Co can be defined by fine changes of external conditions.

As it was shown in literature [11,12,13], the optical band gap energy of TiO_2_ doped with various transition metals (V, Cr, Mn, Fe, Co, Ni, Au, Ag, Pt, Nb, Cu, W, etc.) is lower than in undoped samples. That means an existence of some additional energy levels inside the band gap. This fact was confirmed by ab initio band calculations based on the density functional theory (DFT), namely that a new electron-occupied level occurs, and electrons can be localized around each dopant.

It also should be noted that TiO_2_ crystal structures are highly oxygen defective and can be considered as open to oxygen exchange with external ambient atmosphere. As it was shown such oxygen defects will distort Ti–O octahedron and modified an electronic structure [14,15], and can be revealed by studies of dielectric properties [16].

As shown in literature [17], the stability of anatase phase largely depends on the ionic radius and on the valence of the doping elements. It should be highlighted that photocatalytic performance of anatase polymorph is higher than that of the rutile polymorph. The electronic properties of anatase can be changed by Co doping and consequently can strongly influence the photocatalytic activity of the TiO_2_ systems.

It also should be mentioned that along with attempts to narrowing band gap by doping with metals including Cu and theoretical studies of the results of such doping [18], some new approaches such as band shape engineering were developed [19]. This method results in nanosheet morphology of rutile (110) facets of TiO_2_ doped with Cl, that significantly reduce the effective mass of generated electrons and holes, and consequently enhance photocatalytic activity [19].

Herein, the results of the preparation of Co–doped TiO_2_ nanoparticles as well as the results of the studies of their dielectric properties and spectral dependance of their photocatalytic constant are reported. Degussa P25 powder was chosen because it is usually considered as the “gold” standard for the photocatalysts due to its high photocatalytic performance. This means, it is easier to determine particular influence of doping with Co on the photocatalytic activity. To elucidate the influence of oxygen defects on the electronic structure, the temperature dependence of the dielectric constant along with spectral characteristics of the photocatalytic decomposition constant were studied for different concentrations of Co in the TiO_2_ matrix.

## 2. Materials and Methods

### 2.1. Preparation of Samples

Synthesis of TiO_2_ nanoparticles doped with Co was achieved by precipitation method using the Degussa P25 nanopowder mixed with Co precipitate. As the precipitate was used Tris(diethyldithiocarbamate)Co(III) (CoDtc–Co[C_2_H_5_)_2_NCS_2_]_3_) dissolved in chloroform (CHCl_3_) and then mixed with P25 powder. The resulting compound was dried in air at approximately 50 °C. Five different nominal Co contents in Ti_1-x_Co_x_O_2_ (x = 0.005, 0.01, 0.02, 0.05 and 0.10) were prepared. CoDtc was decomposed by annealing in a furnace at 400° C for 30 min in air. In order to enhance the diffusion of the Co precipitate into the TiO_2_ matrix, the samples were annealed in air at 550 °C during 100 h. Additionally, three different series of sol-gel powders (S1, S2, S3) were prepared by varying the annealing/drying temperature and by varying the order of addition of H_2_O and HCl during the initial stages of preparation. 

### 2.2. Composition and Phase Control

The composition, structure, crystal phases as well as effective surface area of all TiO_2_:Co samples were studied using X-ray Photoelectron Spectroscopy (XSAM800 (KRATOS) X-ray Spectrometer, CQFM, Instituto Superior Técnico, Universidade de Lisboa, Lisbon, [20]), SEM (Zeiss Supra 25 field emission SEM (FESEM) and EDX (EDX detector X-Flash 3001 with Quantax (Bruker AXS, Berlin, Germany), XRD (PANalytical x´PertPRO XRD instrument, equipped with analytic software [21,22], Algarve University, Faro, Portugal,) and Micromeritics ASAP 2020—Physisorption Analyzer (Rostock University, Rostock, Germany). Detailed description of the used analytical methods were published elsewhere [10,16,23].

### 2.3. Photocatalytic Activity Measurements

The photocatalytic activity measurements were carried out on a local-built photocatalytic reactor. The photocatalytic reactor was constructed in cylindrical geometry with a volume of irradiated solution of about 100 mL (with diameter 80 mm and deepness 20 mm), sealed with quartz windows and with a magnetic stirrer inside. As a source of light high-pressure mercury UV-lamp was used. Edmund Optics light filters were used in order to select different spectral lines for solution irradiation, namely 350–380 nm, 401–415 nm, 433–440 nm, 530–565 nm, 565–600 nm and 680–715 nm.

The detailed description of the analytical process for the photocatalytic constant determination was done in [23].

### 2.4. Dielectric Constant Measurements

The dielectric constant (ε) was calculated from the measured capacitance (C), sample thickness (d), free space dielectric permittivity (ε_0_) and the area of the capacitor (A) using the relation ε = (Cd)/(ε_0_A). The capacitance and dielectric losses as a function of temperature was recorded using an Agilent Technology 4284A Precision LCR meter(Algarve University, Faro, Portugal). Samples for dielectric constant measurements were prepared from respective powders by pressing them into pellets with a diameter of 12 mm and a thickness of approximately 1 mm with pressure of about 3000 kg/cm^2^. Capacitor electrodes were deposited with silver glue on both sides of the pellets and electrical contacts were established by gluing 1 micrometer golden wires on these electrodes. Dielectric constant studies were effectuated at frequencies of 1 kHz, 10 kHz and 100 kHz frequencies with an applied voltage of 1 V. The measurements were recorded in a temperature interval from 20 °C to 750 °C in a high vacuum chamber (*p*~10^−5^ mbar). Along with Co–doped TiO_2_ samples pure anatase TiO_2_ samples prepared by Sol–Gel method were also studied.

### 2.5. NMR Analysis and Magnetic Properties 

NMR analysis and studies of magnetic properties of prepared TiO_2_:Co nanopowders were also conducted and are described in detail elsewhere [10].

## 3. Results

### 3.1. Composition Analysis by XPS and EDX

Results of XPS and EDX analysis of the samples prepared by above-described procedure are presented in Table 1.

It should be taken into consideration that XPS analysis provides information on the chemical composition of the surface of the samples (deepness 1–3 atomic layers). Unlike XPS, EDX provides information on the composition of more profound layers on the sample. As it follows from Table 1, Co concentration at the surface is higher and the distribution of Co atoms in the samples is not uniform.

It should be also mentioned that for samples with 10% of Co some amount of metallic Cobalt was detected as it follows from Figure 1.

As it can be seen from Figure 1 the TiO_2_:0.1Co sample has some amount of metallic Co and lower concentration of Co^2+^ species.

### 3.2. XRD Analysis of the Samples

Diffraction pattern in full width for the sample TiO_2_:0.1Co as well as the diffraction peaks of anatase (101) and of rutile (110) of Co–doped TiO_2_ samples are shown in Figure 2, Figure 3 and Figure 4.

No additional peaks (impurity or other Co–Ti–O phases) were detected, which confirm the phase stability of the TiO_2_ system under doping with Co.

As shown in Figure 3 and Figure 4, when the Co concentration increases the pattern peaks turns to be slightly broader indicating lattice disordering. These results confirm the results obtained in [24], that Co ions substitute the Ti sites in the host TiO_2_ matrix. It can also be noted that Co atoms are distributed in anatase as well as in rutile phases.

Heat treatments of the samples in air and in vacuum were effectuated in order to study the stability of the Co–doped TiO_2_ system. The results of XRD phase’s studies after preparation and after heating are shown in Table 2.

From Table 2 it can be seen that for the samples prepared at 550 °C, when the Co content was increased, the Rutile fraction first increase and then decreased and at 10% of Co the relation between Anatase and Rutile phases is almost the same as for the original Degussa P25 powder. For the samples annealed at 800 °C CoTiO_3_ phase was detected. For the samples annealed in vacuum there was not detected full transformation in rutile phase. This behavior seems to indicate that Co acts as inhibitor for phase transformation from phase Anatase to phase of Rutile. It was reported in [17], that Co can be considered as promoter for such transformation. It should be noted, however, that the position of Co between other elements in [17], is close to the boundary line that divided elements as promoters and inhibitors.

### 3.3. Dielectric Properties

The temperature dependence of dielectric constant (the real part of the complex permittivity) for samples prepared by Sol-Gel method and for samples of Degussa P25 doped with Co are shown in Figure 5a–c, (general view) and in Figure 6.

As can be seen from Figure 5a–c and Figure 6, the dielectric constant exhibits two diffusive dielectric anomalies located at ~250 °C and ~700 °C, respectively. These anomalies are observed on heating and are suppressed for Co–doped samples. These behaviors are similar to those of relaxor ferroelectrics [25], and to those that were observed in TiO_2_ thin films under heating in vacuum [16]. Figure 7 shows dielectric losses in the respective samples.

It should be noted the oscillating behavior of the dielectric constant and losses with temperature for the samples with Co content of 5% and 10%. An electrical resistance of the samples is shown in Figure 8.

As seen in Figure 8, there are two steps in resistance changes, one is at temperature about 200 °C, with magnitude of about five orders, and another is in temperature range 400–600 °C, also with magnitude of about 4–5 orders. These changes are irreversible and the residual resistance increased with increasing of Co concentration in the samples. So that for non-doped Sol-Gel TiO_2_ the difference in the resistance is about 10 orders in magnitude and for TiO_2_:10% Co the difference is about 7 orders in magnitude.

### 3.4. Effective BET Surface Area

The detailed description an adsorption experiments was presented in [23]. Several structural features can be deduced from the shape of the isotherm and the conclusions about porous structure can be taken from the type of hysteresis [26,27]. Type IV isotherm was observed for each heat-treated titania powder in this paper. Figure 9 and Figure 10 shows adsorption branches of N2 at 77K and typical pore size distributions.

In Table 3 are shown the calculated specific surface areas and pore volumes and pore areas for the samples doped with Co and for the samples S1, S2 and S3 prepared by Sol-Gel method.

### 3.5. Photocatalytic Activity

Photocatalytic activity of Co–doped TiO_2_ anatase can be estimated by the method established and explained in [23]. As a test substance the pesticide (Fenarimol) was used in order to estimate photocatalytic reaction constant. Fenarimol (Riedel, 99.7%) 5 mg/L solutions were prepared in bidistilled water and was left in contact with the reactor cell in the dark (for 12 h) before irradiation in order to achieve the adsorption equilibrium. Briefly, three different degradation processes can occur in the photocatalytic reactor simultaneously. The first process is the degradation on the anatase phase surface and the second is degradation on the rutile phase surface. The third process is the direct decomposition under UV light. The kinetic constant of can be calculated from the first order kinetic Equation (1)
(1)$$\ln\left(\frac{I_{t}}{I_{0}}\right)=-k\times t$$
where *I_t_* and *I*_0_ are the current and initial intensities at 220 nm of the HPLC chromatographic integrated peaks, *k* is the kinetic constant and *t* is time in hours. As it was shown in [27] all three processes of degradation are independent on each other. Consequently, kinetic constant can be represented as a sum of the constants *k* = *k_A_* + *k_R_* + *k_D_*, where *k_A_*, *k_R_*, are the constants for anatase and rutile phases and *k_D_* is the constant of direct degradation. Using data from the Table 2 where the fractions of the anatase and rutile phases are present for each sample, it is possible to calculate the specific kinetic constant for the anatase phase, according to Equation (2) [23]:(2)$$k_{A}=\frac{k-\left(m_{R}\times k_{R}+k_{D}\right)}{m_{A}}$$

Taking into account the data in Table 3, where the specific area for the Co–doped TiO_2_ samples is shown, it is possible to calculate the specific kinetic constants, normalized to the surface area, in according with Equation (3).
(3)$$k_{AS}=\frac{k_{A}}{\mathrm{Specific}\;\mathrm{area}}$$

The results are presented in Table 4.

As it is followed from Table 4, a strong decrease in photocatalytic activity was observed when Co content increased. So that for the P25:10%Co samples the photocatalytic activity is almost completely being suppressed. It should be taken into account that BET specific area is not changed drastically as well as Anatase to Rutile ratio (see Table 2 and Table 3). It means that probably the photocatalytic activity is suppressed by electron–hole recombination on Co impurity centres. However, it should be mentioned that Table 4 reflects photocatalytic processes that take place when the complete light spectrum of the high-pressure mercury lamp illuminates the surface of the photocatalysts. Figure 11 shows the spectrum of emitted light of the mercury lamp along with the spectra after optical filters are inserted.

The results of the study of the spectral dependence of photocatalytic constant are represented in Table 5.

It follows from Table 5, that for P25 Degussa powders the photocatalytic constant is 1.53 h^−1^, however, it is 0.26 h^−1^, 0.21 h^−1^, 0.24 h^−1^ and 0.0816 h^−1^ for P25:x%Co with different concentrations of Co (x = 0.5, 1, 2 and 10), respectively.

The obtained values of the photocatalytic constant of sol-gel powders were 0.39 h^−1^, 0.15 h^−1^, and 0.19 h^−1^ for the samples S1, S2 and S3 respectively.

These samples have different surface areas, namely 6.4 m^2^/g, 73.5 m^2^/g and 8.6 m^2^/g for S1, S2 and S3 compared to the P25 Degussa that was 57 m^2^/g.

In addition, these samples have different crystalline sizes e.g., 24.4 nm, 10.75 nm and 33.98 nm for S1, S2 and S3 compared to the 25.6 nm of P25 Degussa.

The efficiency of the photocatalytic activity is mainly affected by the crystalline quality, the surface area and the crystallite sizes of the powdered samples prepared by the sol-gel method.

The photocatalytic constant is lower in these powders when compared to P25 Degussa, probably due to a lower crystalline quality (P25 Degussa powders were prepared at significantly higher temperatures) and to the coexisting anatase and rutile crystal modifications. This conclusion can be done because of both surface area and crystalline size of Sol-Gel powders and P25 Degussa are of the same order.

Furthermore, the sample S1 powder showed the highest photocatalytic constant compared to the series S2 and S3 powders, respectively.

On the other hand, spectral dependence of the photocatalytic constant was studied for TiO_2_ nanopowders including P25 Degussa, sol-gel samples and P25 Degussa doped with different concentrations of Cobalt. As it can be seen in Table 5, some photocatalytic activity can be observed for all nanopowders.

## 4. Discussion

As it can be seen from Figure 5a–c and Figure 6 there are two distinct temperature intervals for temperature dependence of dielectric constant, for dielectric losses and for the resistance of the samples, namely temperatures less than 400 °C and higher temperatures. At temperatures higher than 400 °C, anomalies of dielectric constant (ε(T)) in dependence on temperature were observed for all samples under heating in vacuum including the samples prepared by Sol-Gel method. However, these anomalies were suppressed when the Co concentration is increased. Comparing these results with results obtained in studies of thin films [16], it can be concluded that the mechanism which is responsible for this compartment of dielectric susceptibility is likely the same. Namely, it can be related with the reorientation alignment of the oxygen—defected octahedral. In this process defects created by Co atoms incorporated in TiO_2_ matrix will restrict the correlation distance and as a result suppress the peak of dielectric constant anomaly.

Instability of dielectric constant and dielectric losses at temperatures below 400 °C have essentially lower amplitude and different character for doped and for non-doped samples. In addition, for Co doped samples some oscillations of ε(T) and ε″(T) in dependence of the temperature were observed. It can be seen that the peak of the anomaly of ε(T) is clearly observed at about 200 °C and it is suppressed when the Co concentration is about 10%. Dielectric losses (ε″(T)) have another anomalous behavior at temperatures above 300 °C. From Figure 6 it can be noted that changes in the electrical resistance R(T) start at 200 °C. So, the dielectric anomalies are related with changes in the electronic structure of TiO_2_ powders.

As it is follows from Table 5, for all spectral lines presented in the mercury lamp light, photocatalytic activity of the nanopowders was observed. Detailed analysis shows the presence of electronic states inside the main energy gap between valence and conduction bands of TiO_2_. These states can produce electron–hole pair under radiation with different wavelengths higher than the wavelength of the corresponding energy gap.

It can be seen from Table 5 that for the λ = 365 nm, the normalized photocatalytic constant k is higher for P25 Degussa (*k* = 885) powder, but with increasing Co concentration the k increased from about *k* = 117 to *k* = 462 when Co concentration changed from 1% to 10%. For λ = 405 nm, and for the sample P25:5%Co, k is higher (*k* = 683) than for the P25 Degussa (*k* = 250). The same was observed for λ = 436 nm. For λ = 546 nm all values of k are relatively low. Finally, for λ = 690 nm, it was observed relatively high values of k for all samples doped with Co as well as for P25 Degussa sample.

This behavior of k can be interpreted in terms of the existence of additional energy levels inside the main energy gap of TiO_2_, and that these levels can be changed by changing of Co concentration.

## 5. Conclusions

In conclusion, the possibility of preparation of Co–doped TiO_2_ samples via solid state synthesis of a mixture of Degussa P25 nanopowder with Co precipitate was demonstrated. In according with the XRD data the Co–doped TiO_2_ powders shows the formation of the anatase and rutile phases at 550 °C without any indication of the presence of metallic dopants within the sensitivity of XRD method. It is necessary to mention, however, that by XPS method some small amount of metallic Co was detected. This probably indicates the limit of Co dissolution in the TiO_2_ matrix. Heating of the Co doped TiO_2_ samples in air and in vacuum up to 750–800 °C results in CoTiO_3_ phase formation. This indicates instability of the uniform distribution of cobalt in TiO_2_ lattice under heating. The study of temperature dependence of dielectric properties and electrical resistance of Co–doped TiO_2_ samples revealed very large changes in the dielectric constant as well as in resistance, when measured in situ during heating in vacuum. These changes are irreversible and can be assigned to the generation of electrical charge carriers due to oxygen vacancies formation, during heating in vacuum. As shown earlier from studies of magnetic properties of Co doped TiO_2_ samples [10], saturated magnetic moment indicates the presence of different oxidation states of cobalt that depend of the content of this element. It has been suggested that oxygen vacancies are essential to provide the s–d exchange coupling between cobalt ions leading to AFM state [10]. The photocatalytic activity under full-spectrum irradiation from the light of a high-pressure mercury lamp shows suppression of this activity when the cobalt content increased, despite the energy gap and BET surface area being almost the same. This fact indicates that electron–hole recombination time is probably much shorter then in non-doped TiO_2_. It can then be assumed that the following factors are essential for photocatalytic activity of TiO_2_:Co nanopowders. First, atoms of Co incorporated in the TiO_2_ matrix create additional electronic states inside the TiO_2_ energy gap, which results in enhanced photocatalytic activity at wavelengths λ ≥ 400nm. Second, some additional electronic levels inside the energy gap are also created by crystal defects and reveal itself by photocatalytic activity in nanopowders without the presence of any dopants. Third, s–d interaction between Co atoms that results in AFM interaction [10,28] also results in faster electron–hole recombination. These results are in consistence with studies of black (conductive) TiO_2_ powders prepared by annealing in hydrogen reduced atmosphere [29] and with red TiO_2_ described in [15].

## Figures and Tables

**Figure 1 nanomaterials-11-02519-f001:**
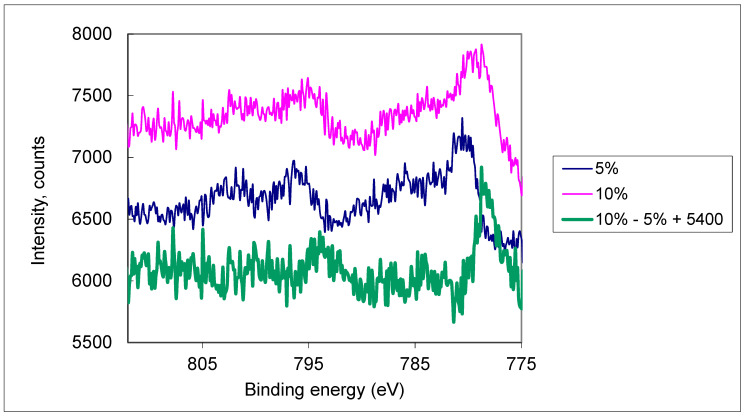
XPS spectra of the samples TiO_2_:0.05Co and TiO_2_:0.1Co.

**Figure 2 nanomaterials-11-02519-f002:**
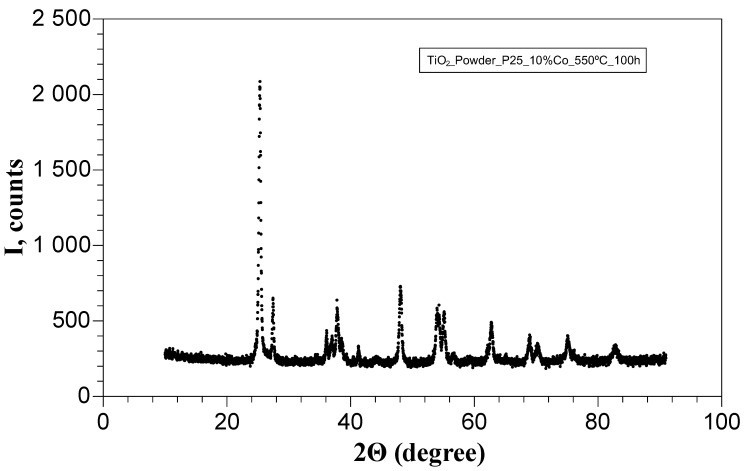
XRD Diffraction pattern of the sample TiO_2_:0.1Co.

**Figure 3 nanomaterials-11-02519-f003:**
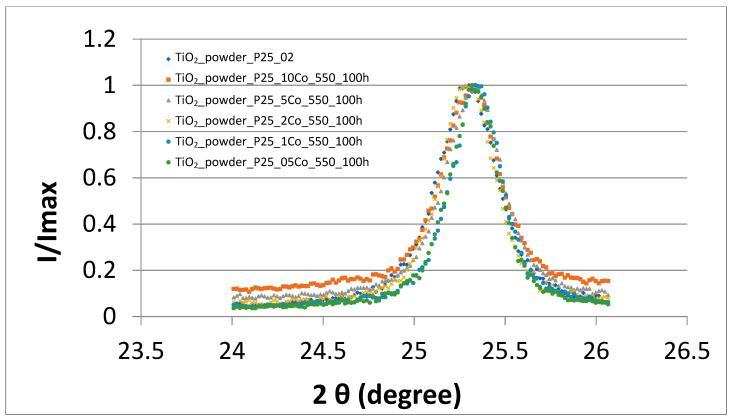
XRD peak ((101)—anatase) patterns of TiO_2_ Degussa P25 powders doped with Co.

**Figure 4 nanomaterials-11-02519-f004:**
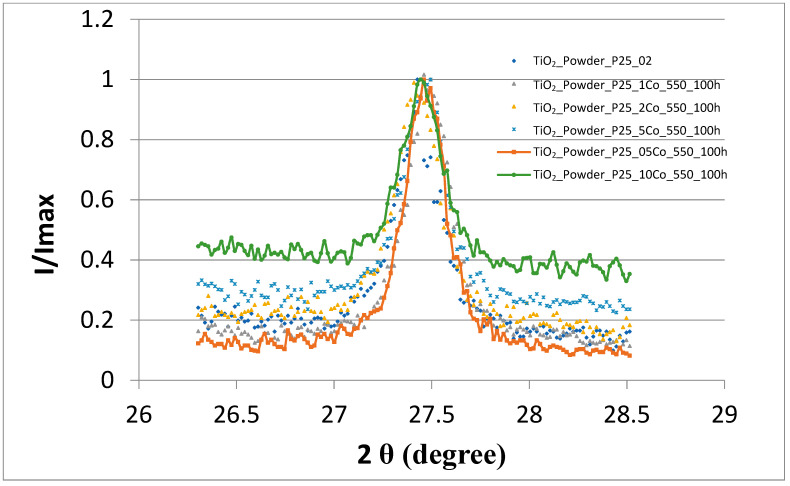
XRD peak ((110)—rutile) patterns of TiO_2_ Degussa P25 powders doped with Co.

**Figure 5 nanomaterials-11-02519-f005:**
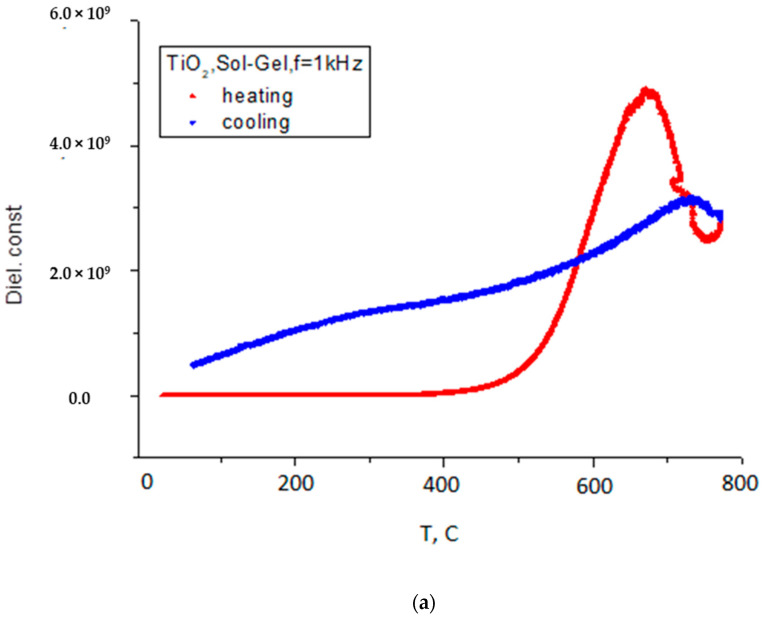
(**a**) Dielectric constant of pure anatase TiO_2_ sample prepared by Sol-Gel method. (**b**) Dielectric constant of Degussa P25 TiO_2_ sample doped with 2% Co. (**c**) Dielectric constant of Degussa P25 TiO_2_ sample doped with 10% Co.

**Figure 6 nanomaterials-11-02519-f006:**
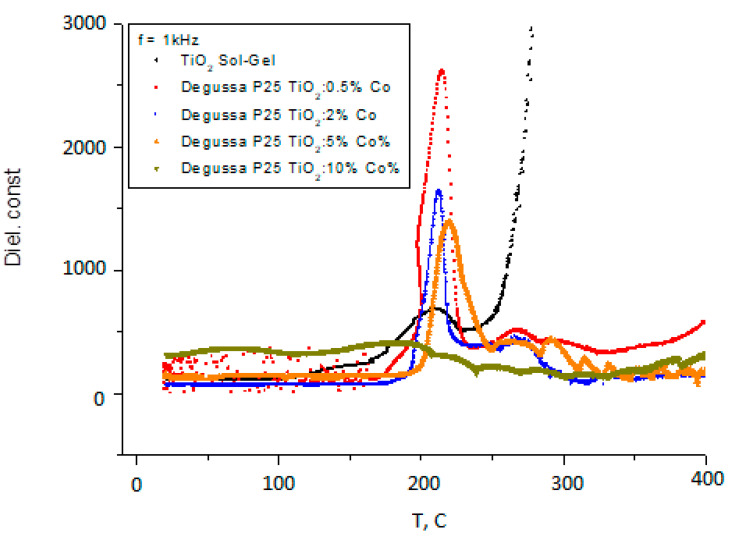
Dielectric constant of Degussa P25 TiO_2_ sample doped with different contents of Co.

**Figure 7 nanomaterials-11-02519-f007:**
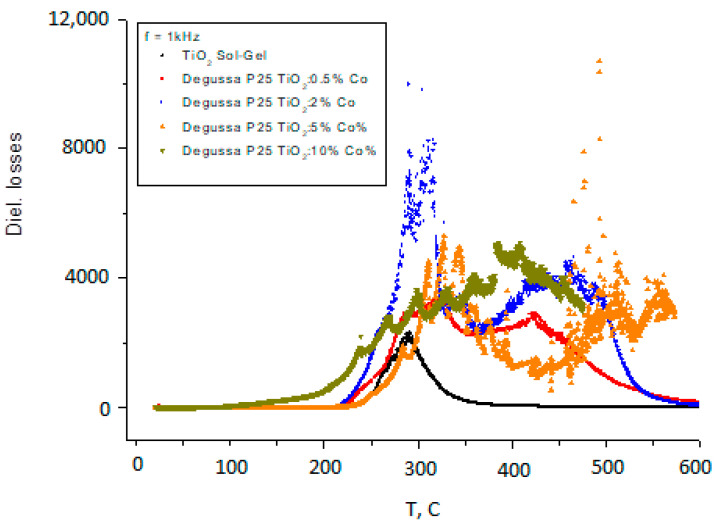
Dielectric losses of Degussa P25 TiO_2_ sample doped with different contents of Co.

**Figure 8 nanomaterials-11-02519-f008:**
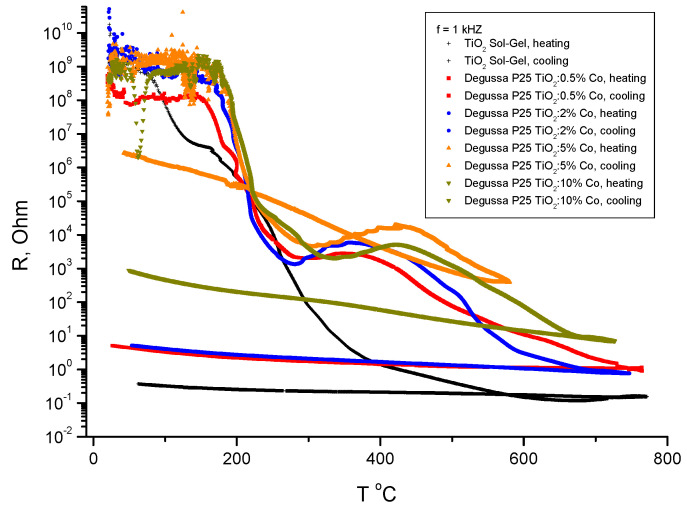
Electrical resistance of the Co–doped TiO_2_ samples on heating and on cooling.

**Figure 9 nanomaterials-11-02519-f009:**
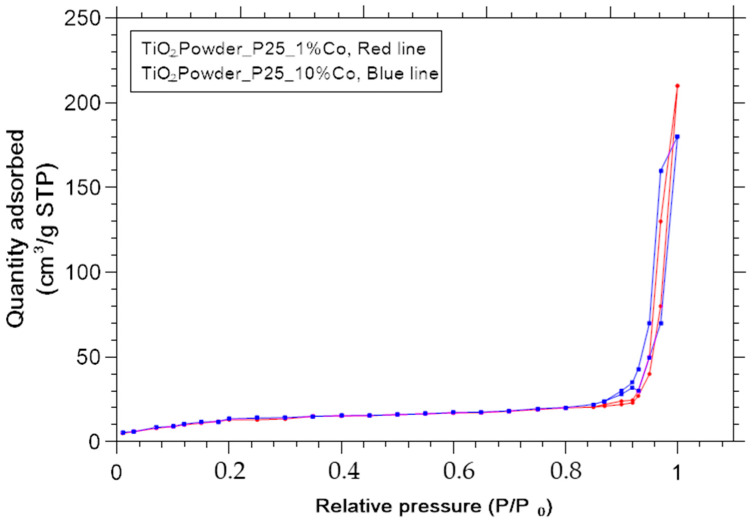
Adsorption branches of N2-Adsorption measurements at 77K.

**Figure 10 nanomaterials-11-02519-f010:**
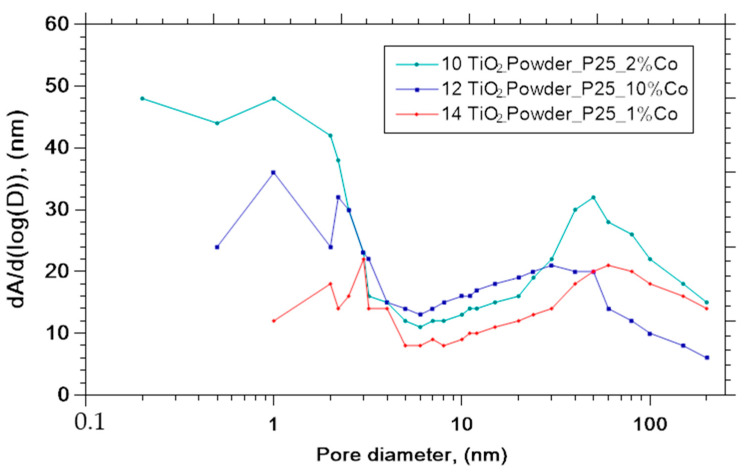
Pore size distribution of selected samples.

**Figure 11 nanomaterials-11-02519-f011:**
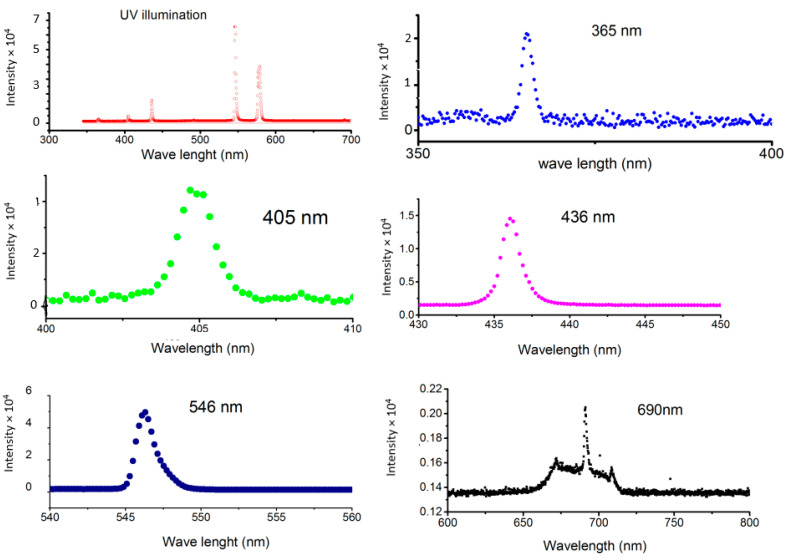
Spectrum of high-pressure mercury lamp and spectra of different spectral lines after the optical filters.

**Table 1 nanomaterials-11-02519-t001:** Results of composition analysis by XPS and EDX of Co–doped TiO_2_ samples.

TiO_2_:Co		XPS	EDX
Co (%)	Co/Ti	Co/Ti	Co/Ti
0.5	0.005	0.032	0.033
1	0.01	0.119	0.042
2	0.02	0.115	0.059
5	0.05	0.144	0.128
10	0.1	0.211	0.213

**Table 2 nanomaterials-11-02519-t002:** XRD analysis of the TiO_2_:Co samples prepared at 550 °C, annealed at 800 °C during 24 h and heated in vacuum up to 750 °C.

Sample Prepared at 550 °C			Samples Annealed in Air at 800 °C, 24 h		Samples Heated in Vacuum up to 750 °C	
Samples Ti_1−x_Co_x_O_2_ %, of Co	Anatase %, of Fraction	Rutile %, of Fraction	Rutile %, of Fraction	CoTiO_3_ %, of Fraction	Rutile %, of Fraction	CoTiO_3_ %, of Fraction
0	83.3	16.7	100	0		
0.5	72.6	27.4	99.4	0.6		
1	75.6	24.4	99.2	0.8		
2	78.0	22.0	97.0	3.0		
5	80.1	19.9	91.3	8.7		
10	83.0	17.0	82.5	17.5	86.7	13.3

**Table 3 nanomaterials-11-02519-t003:** BET specific surface areas and pore areas.

Co Content, %	BET Surface Area (SA), m^2^/g	SA Error, m^2^/g	Pore Volume, cm^3^/g	Pore Volume Error, cm^3^/g	Pore Area, m^2^/g
0, Degussa P25	41.27	1.3			
0, P25 Annealed at 550 °C in air	44.9	1.3			
0	28.9	1.3	0.231	0.01	25.82
0.5	31	1.3	0.273	0.011	30.91
1	31.5	1.3	0.325	0.014	27.10
2	38	1.6	0.420	0.017	42.94
5	35.5	1.5	0.307	0.013	34.68
10	31.5	1.3	0.270	0.011	33.98
S1	6.4	0.1	0.01	0.01	6.8
S2	73.5	1.1	0.08	0.09	75.4
S3	8.6	0.12	0.07	0.01	9.8

**Table 4 nanomaterials-11-02519-t004:** Photocatalytic activity of the Co–doped TiO_2_ samples.

Composition	*k*, h^−1^	*k_A_*/g, (h × g)^−1^	*k_AS_*/m^2^, (h × m^2^)^−1^
Direct degradation	0.058		
Degussa P25 (Rutile)	0.1228		
Degussa P25	1.53	71.49	1.732
Degussa P25:0.5%Co	0.2664	24.07	0.776
Degussa P25:1%Co	0.21	18.97	0.700
Degussa P25:2%Co	0.2441	20.39	0.537
Degussa P25:10%Co	0.0846	0.69	0.022
S1	0.39	35.24	5.5
S2	0.15	13.55	0.17
S3	0.19	17.17	1.99

**Table 5 nanomaterials-11-02519-t005:** Photocatalytic constant *k* of the studied samples normalized to the integral intensity of the spectral peaks for different wavelength emitted by mercury lamp. In the Table are represented values of *k* × 10^7^.

Wavelength	365 nm	405 nm	436 nm	546 nm	690 nm
P25 Degussa	885	250	12.1	2.97	420
P25:1%Co	117	76.8	12.1	1.99	245
P25:2%Co	110	327	8.31	0.83	271
P25:5%Co	175	683	104	4.05	245
P25:10%Co	462	65	9.1	1.2	17
S1	72.9	42.3	7.67	2.35	16.6
S2	45.8	21.1	12.6	2.87	21.01
S3	188	53.8	7.03	0.88	15.7

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
