# Peer review of "Dielectric Properties and Spectral Characteristics of Photocatalytic Constant of TiO2 Nanoparticles Doped with Cobalt"

_nanomaterials, 2021, doi:10.3390/nano11102519_

Round 1

Reviewer 1 Report

In the manuscript, the authors reported the dielectric properties and spectral dependence of the photocatalytic constant of Co doped 23 P25 Degussa powder, and its the structures were characterized well. Add the explanation of the photocatalytic reaction and improve the quality of Figures (especially Figure 2). Introduction needs to add new methods.  Overall, the manuscript is well organized, and can be published in the present form.

Author Response

Dear Referees,

First of all we are expressing our gratitude to referees for their work trying to improve our manuscript.

Please find our revised manuscript “Dielectric properties and spectral characteristics of photocatalytic constant of TiO2 nanoparticles doped with cobalt” prepared by V.G. Bessergenev*, J.F.M.L. Mariano, M.C. Mateus, J.P. Lourenço, Adwaa Ahmed, M. Hantusch, E. Burkel and A.M. Botelho do Rego.

We changed our manuscript in according with referees suggestions:

First, the quality of some Figures was improved.

Second, Figure 2 that reflects XRD pattern in full width was included in the manuscript.

Third, In the Introduction the part that reflects a new approach of band shape engineering has been included along with respective references. 

Forth, we have to mention that enlargement of the diffraction peaks of (101) for anatase and (110) for rutile confirms that both crystalline phases were doped with Co.

Fifth, BJH plots reflect that morphology of the doped samples did not change significantly and that changes in photocatalytic activity can be attributed to the modifications in electronic structure of the samples. So, these plots should be included in the manuscript in order to clarify the situation. 

Sixth, the part of the text explaining the photocatalytic reaction was added in the manuscript.

With my best regards,Prof. Dr. Valentin Bessergenev

17.09.2021

Reviewer 2 Report

The authors have studied the Co doping effect on P25 in photocatalysis and the dielectric constant. This type of research is a library and fundamental research and has the potential to be interesting for the readership. However, the manuscript has some critical matters to revise and before that, it is not possible to accept it. Some of the main issues are below:

-The figures should be drawn in acceptable quality. some of them are of very low quality. All figures should be drawn like Figure 10. 

-Why the authors have not shown any XRD in full width? The full width is very important and should be added to the manuscript. 

-BJH plot is only a theoretical calculation for mesoporous materials, therefore, it should be removed from the manuscript. 

-What is the photocatalytic reaction? it is not clearly presented in the manuscript.

-The introduction is weakly written, the dopants have not only an effect on the band gap but also on the band structure. This is never discussed in the context of the manuscript. For more information, the authors can use the following paper and cite it in the manuscript: Applied Catalysis B: Environmental, 2021, 297, 120380.

Furthermore, it is difficult to say which phase has been influenced more by doping, anatase, or rutile? Even the planes have different interactions with the dopant, for such studies, monophase is more suitable. See the reference: Applied Surface Science, 2016, 387, 682-689

Furthermore, the applications and the importance of P25 in and the TiO2 heterostructures should be mentioned in the catalytic, photocatalytic, and environmental applications, for this, the following papers are useful: 

"CuO-NiO-TiO2 bimetallic nanocomposites for catalytic applications." Molecular Catalysis 496 (2020): 111193.

Desalination, 393, 65-78. DOI: 10.1016/j.desal.2015.07.003

Fundamentals of TiO2 photocatalysis: concepts, mechanisms, and challenges. Advanced Materials31(50), 1901997

Journal of Membrane Science, 489, 43-54. DOI: 10.1016/j.memsci.2015.04.010

Desalination, 292, 19-29. DOI: 10.1016/j.desal.2012.02.006

Author Response

Dear Editor,
First of all we are expressing our gratitude to referees for their work trying to improve our manuscript.
Please find our revised manuscript “Dielectric properties and spectral characteristics of photocatalytic constant of TiO2 nanoparticles doped with cobalt” prepared by V.G. Bessergenev*, J.F.M.L. Mariano, M.C. Mateus, J.P. Lourenço, Adwaa Ahmed, M. Hantusch, E. Burkel and A.M. Botelho do Rego.
We changed our manuscript in according with referees suggestions:
“Question: -The figures should be drawn in acceptable quality. some of them are of very low quality. All figures should be drawn like Figure 10.”
Response: The quality of some Figures was improved.
“Question: -Why the authors have not shown any XRD in full width? The full width is very important and should be added to the manuscript.”
Response: Figure 2 that reflects XRD pattern in full width was included in the manuscript.
“Question: -BJH plot is only a theoretical calculation for mesoporous materials, therefore, it should be removed from the manuscript.”
Response: BJH plots reflect that morphology of the doped samples did not change significantly and that changes in photocatalytic activity can be attributed to the modifications in electronic structure of the samples. So, these plots should be included in the manuscript in order to clarify the situation.
“Question: -What is the photocatalytic reaction? it is not clearly presented in the manuscript.”
Response: The part of the text explaining the photocatalytic reaction was added in the manuscript.
“Question: -The introduction is weakly written, the dopants have not only an effect on the band gap but also on the band structure. This is never discussed in the context of the manuscript. For more information, the authors can use the following paper and cite it in the manuscript: Applied Catalysis B: Environmental, 2021, 297, 120380.”
Response: In the Introduction the part that reflects a new approach of band shape engineering has been included along with respective references.
“Question: Furthermore, it is difficult to say which phase has been influenced more by doping, anatase, or rutile? Even the planes have different interactions with the dopant, for such studies, monophase is more suitable. See the reference: Applied Surface Science, 2016, 387, 682-689.”
Response: An enlargement of the diffraction peaks of (101) for anatase and (110) for rutile confirms that both crystalline phases were doped with Co. Respective reference: Applied Surface Science, 2016, 387, 682-689, was added to the manuscript.
“Question: Furthermore, the applications and the importance of P25 in and the TiO2 heterostructures should be mentioned in the catalytic, photocatalytic, and environmental applications, for this, the following papers are useful:
"CuO-NiO-TiO2 bimetallic nanocomposites for catalytic applications." Molecular Catalysis 496 (2020): 111193.
Desalination, 393, 65-78. DOI: 10.1016/j.desal.2015.07.003
Fundamentals of TiO2 photocatalysis: concepts, mechanisms, and challenges. Advanced Materials, 31(50), 1901997
Journal of Membrane Science, 489, 43-54. DOI: 10.1016/j.memsci.2015.04.010
Desalination, 292, 19-29. DOI: 10.1016/j.desal.2012.02.006”
Response: In the Introduction an importance of TiO2 powders, thin films and heterostructures for various applications, including photocatalysis are already mentioned, see references 1-7.
With my best regards,
Prof. Dr. Valentin Bessergenev
17.09.2021

Round 2

Reviewer 2 Report

The authors have replied to my concerns and comments and it is acceptable in the current format for publication.

This manuscript is a resubmission of an earlier submission. The following is a list of the peer review reports and author responses from that submission.